# Insight into the Double-Edged Role of Ferroptosis in Disease

**DOI:** 10.3390/biom11121790

**Published:** 2021-11-30

**Authors:** Lei Zhang, Ruohan Jia, Huizhen Li, Huarun Yu, Keke Ren, Shuangshuang Jia, Yanzhang Li, Qun Wang

**Affiliations:** 1Henan International Joint Laboratory for Nuclear Protein Regulation, Henan University, Kaifeng 475004, China; zhlei@henu.edu.cn (L.Z.); 1819010242@vip.henu.edu.cn (R.J.); 1819010278@vip.henu.edu.cn (H.L.); 2School of Basic Medical Sciences, Henan University, Kaifeng 475004, China; 104753190848@henu.edu.cn; 3School of Clinical Medicine, Henan University, Kaifeng 475004, China; 1922010157@henu.edu.cn (H.Y.); 1922010171@henu.edu.cn (K.R.)

**Keywords:** ferroptosis, systemic diseases, ROS, inhibitors, inducers

## Abstract

Ferroptosis, a newly described type of iron-dependent programmed cell death that is distinct from apoptosis, necroptosis, and other types of cell death, is involved in lipid peroxidation (LP), reactive oxygen species (ROS) production, and mitochondrial dysfunction. Accumulating evidence has highlighted vital roles for ferroptosis in multiple diseases, including acute kidney injury, cancer, hepatic fibrosis, Parkinson’s disease, and Alzheimer’s disease. Therefore, ferroptosis has become one of the research hotspots for disease treatment and attracted extensive attention in recent years. This review mainly summarizes the relationship between ferroptosis and various diseases classified by the system, including the urinary system, digestive system, respiratory system, nervous system. In addition, the role and molecular mechanism of multiple inhibitors and inducers for ferroptosis are further elucidated. A deeper understanding of the relationship between ferroptosis and multiple diseases may provide new strategies for researching diseases and drug development based on ferroptosis.

## 1. Introduction

In the past, apoptosis, a programmed cell death, and necrosis, which is a passive death, were thought to be two different forms of cell death. Characterized by some characteristic morphological changes in cell structure and some enzyme-dependent biochemical processes, apoptosis results in clearing cells from the body with minimal damage to surrounding tissue. However, necrosis usually manifests itself as uncontrolled cell death, usually after a severe insult that causes the contents of the cell to spill over into surrounding tissues and subsequently causes damage. In recent years, more cell death modes, such as pyroptosis, autophagy, and ferroptosis have been discovered. Ferroptosis is a new cell death mode first reported by Dixon et al., in 2012 [1]. Ferroptosis is an iron-dependent cell death characterized by the decrease in glutathione (GSH) peroxidase 4 (GPX4) activity and deposition of lipid peroxide, which is different from other cell death modes in biochemistry, morphology, and heredity [2]. In morphology, the density of mitochondria fell, the cristae of mitochondria decreased or disappeared, the outer membrane of mitochondria is broken, but the cell membrane remains intact, the nuclear size is standard, and the chromatin is not condensed [3]. At the biochemical level, ferroptosis is mainly caused by the abnormal increase in ROS mediated by the accumulation of iron-dependent lipid peroxides.

The metabolic pathways of ferroptosis mainly involve the iron-ion metabolism pathway and the ROS metabolism pathway. The iron metabolism pathway is that in which Fe^3+^ is reduced to Fe^2+^ in the endosome by binding to transferrin receptor 1 (TFR1), and Fe^2+^ is released into the cytoplasm through divalent metal transporter 1 (DMT1), which can initiate LP and produce ROS through the Fenton reaction, thus leading to ferroptosis. Intracellular excess iron is stored in ferritin [4].

Iron-dependent ROS metabolism pathways include the cystine/glutamate transporter (System Xc-) pathway [5], GPX4 [6] inactivation based on GSH consumption, and mitochondrial voltage-dependent anion channels (VDACs). As a membrane-type Na^+^-dependent Cys-Glu exchange transporter, System Xc-, which consists of SLC7A11 and SLC3A2, absorb Cys and excrete Glu. System Xc- is introduced into cystine by a disulfide heterodimer encoded by the SLC7A11 gene to maintain homeostasis of REDOX, and then reduced to cysteine for the synthesis of glutathione [7]. Erastin inhibits System Xc- and depletes GSH, reducing GPX4 activity and leading to lipid peroxides accumulation, which further leads to an increase in ROS. Skonieczna et al. found that erastin is one of the targets of VDACs, which can reduce the oxidation of Nicotinamide adenine dinucleotide (NADH) by closing the VDAC2 and VDAC3 channels, leading to the decrease in GSH content and further leading to ferroptosis [8]. RAS-selective lethal small molecule 3 (RSL3), diphenyleneiodonium chloride (DPI)7 [9], DPI10, etc., are ferroptosis inducers, which can directly act on GPX4 and inhibit its activity, leading to LP intracellular, ROS generation, thereby causing cell ferroptosis [10].

In recent years, some studies have shown that ferroptosis is closely related to various human diseases, such as colorectal cancer (CRC), prostate cancer (PCA), Alzheimer’s disease (AD), Parkinson’s disease (PD), and acute renal failure (Table 1). Therefore, to understand the main characteristics and regulatory mechanisms of ferroptosis, as well as the influencing pathways and nodes of iron metabolism in vivo on the occurrence of ferroptosis, it is beneficial to understand the correlation between clinical diseases and ferroptosis to provide new ideas for the mechanism research of disease occurrence and clinical drug targets.

## 2. Diseases

### 2.1. Urinary System

Urinary system diseases are common clinical diseases. According to data, 1,200,000 people died from chronic kidney disease (CKD) in 2017, matching the number of people who die from all urinary diseases [11]. Abnormal activation of ferroptosis is closely associated with the development of acute kidney injury, while mis-inhibition of ferroptosis is associated with the development of prostate cancer. We review the friend-versus-foe relationship between urinary diseases and ferroptosis through the following parts: AKI, PKD, and PCA (Table 1).

#### 2.1.1. Acute Kidney Injury (AKI)

AKI is considered to be a systemic disease, and it has been traditionally thought that the pathogenesis of AKI may be related to vasoconstriction, oxidative stress, apoptosis, and inflammation [12,13]. Therefore, the elucidation of more detailed mechanisms regarding the occurrence of AKI is of great importance for the clinical treatment of AKI. Interestingly, since ferroptosis was reported in 2012 [1], many investigations have indicated that ferroptosis is connected to AKI. At the molecular biology level, single-cell RNA sequencing (sc-RNA-seq) showed that the expression of the ferroptosis-related gene ACSL4 was up-regulated in tubular epithelial cells, while necrosis- and scorch death-related genes were less expressed in this group of cells [14]. Currently, hu et al. found that the activation of vitamin D receptor (VDR) can inversely regulate GPX4, thereby inhibiting ferroptosis and cisplatin-induced AKI [15]. Heme oxygenase (HO) is the rate-limiting enzyme in the catabolism of heme, and its abnormal expression is associated with the occurrence of ferroptosis [16]. Adedoyin O et al. claimed that their studies demonstrate ferroptosis can drive cell death in the renal proximal tubule. In contrast, HO-1 expression indicates an antiferroptotic effect, thereby inhibiting AKI Pannexin 1 (PANX1, an ATP-releasing pathway family protein), which has pro-apoptotic effects during kidney injury [17]. Su et al. demonstrated that PANX1 deletion induces the expression of HO-1 and inhibits ferroptinophagy via the mitogen-activated protein kinase (MAPK)/extracellular signal-regulated kinase (ERK) pathway, which protects against AKI by regulating ferroptosis [18]. Furthermore, Guo et al. found that Rev-erb-α/β (circadian clock component) promotes ferroptosis by directly binding to RORE (a motif for Rev-erb binding) cis-elements and repressing the transcription of genes encoding SLC7A11 and HO-1 (two proteins that inhibit ferroptosis) [19]. Ferroptosis plays a contributing role in the development of AKI. Therefore, the inhibition of ferroptosis is crucial in the fight against AKI, and the discovery of an increasing number of compounds and biological enzymes related to it has made this possible. It is believed that as research proceeds, it will eventually be possible to clinically inhibit the “enemy” ferroptosis to combat AKI.

#### 2.1.2. Chronic Kidney Disease and Renal Tumors

Polycystic kidney disease (PKD), CKD, and Renal cell cancer (RCC) are kidney diseases that have been associated with ferroptosis. In PKD, ferroptosis may act as a non-specific pure damage-associated molecular pattern release mechanism, or it may lead to peroxidation of lipid-specific neoepitopes [20]. In the conversion pathway of AKI to CKD, synchronized regulated necrosis may promote nephron loss and contribute to the AKI-to-CKD transition. This process is triggered by necroptosis and executed by ferroptosis, a combination therapy employing necrostatins and ferrostatins hypothesized to protect from irreversible nephron loss Toll-like receptors, Nod-like receptors, and pyrin domain-containing protein 3 (NLRP3) inflammasome, playing an essential role in the pathogenesis of kidney diseases, such as ischemia-reperfusion injury, sepsis-induced AKI, diabetic nephropathy, and unilateral ureter obstruction [20,21,22]. High mobility group box 1, a prototypic damage-associated molecular pattern molecule, was demonstrated as a mediator of ferroptosis-induced inflammation [20,23]. RCC mostly shows resistance to chemotherapy and radiotherapy treatment [20]. However, tumor cells not only enhance oxygen-consuming metabolic activities, but also resist oxidative stress damage [24]. Therefore, A therapeutic strategy targeting metabolic processes may be a better choice for treating cancer [24]. Yang et al. tested the sensitivity of 117 cancer cell lines to erastin-induced ferroptotsis and revealed that diffuse large B cell lymphomas and renal cell cancers are particularly susceptible to GPX4-regulated ferroptosis [9]. Another study indicated that the Hippo pathway effector TAZ regulates ferroptosis sensitivity in RCC by improving the level of Epithelial Membrane Protein 1 and inducing the expression of nicotinamide adenine dinucleotide phosphate (NADPH) Oxidase 4 (NOX4) [25]. Above all, we know understanding the different mechanisms may provide new ideas in treating kidney cancer in clinics.

#### 2.1.3. Prostate Cancer (PCA)

According to the National Cancer Institute, prostate cancer (PCA) is the most common cancer in North American men, excluding skin cancer [26]. Unlike AKI, in PCA, ferroptosis is inhibited, and modulating the target of ferroptosis and activating ferroptosis may be an effective approach to treat advanced PCA. The accumulation of ROS is thought to be a key step in the development of ferroptosis. ATP production in cells is accompanied by the accumulation of ROS, while under starvation conditions, mitochondrial activity is reduced, and ROS production decreases. Ogor p et al. find that VCP (the most abundant soluble ATPase) in starvation-treated PCA cells can self-aggregate to help cancer cells resist death, and this process may also have some connection to ferroptosis, but particular mechanisms remain to be explored [27]. Tousignant KD et al. corroborate that the rise in membrane PUFA (iron-dependent peroxidation of polyunsaturated fatty acids) levels enhances membrane fluidity and LP, which causes hypersensitivity to GPX4 inhibition and ferroptosis [28]. Nassar ZD et al. substantiate that DECR1 (an androgen-repressed survival factor) causes cellular accumulation of PUFAs, enhances mitochondrial oxidative stress and LP, and induces ferroptosis [29]. Blomme A et al. verify that DECR1 participates in redox homeostasis by regulating the balance between saturated and unsaturated phospholipids and DECR1 knockout induces ER stress and sensitizes PCA cells to ferroptosis [30]. In clinical applications, flubendazole, an anthelmintic drug approved by the FDA, has been reported to produce effective antitumor effects by targeting P53 and promoting ferroptosis formation in castration-resistant PCA [31]. Recently, it has also been reported that LncRNA OIP5-AS1 inhibits ferroptosis in PCA with long-term cadmium exposure through regulating the expression of SLC7A11 [32]. The PI3K-AKT-mTOR signaling pathway was reported to reduce PCA sensitivity to ferroptosis [33]. However, the NF2-Hippo signaling pathway enhanced PCA sensitivity to ferroptosis [33,34]. In the future, it might be possible to treat PCA by modulating these two pathways.

### 2.2. Digestive System

Digestive system tumors have been clinically characterized by high morbidity and mortality [35], and cardia and noncardia gastric cancer alone are responsible for over 1,000,000 new cases in 2018 and an estimated 783,000 deaths [36]. Several molecules, including tumor protein p53, perilipin2, retinoblastoma protein, nuclear factor NRF2, KH RNA binding domain containing signal transduction associated 1, cysteine dioxygenase type 1 (CDO1), metallothionein-1G, nuclear receptor coactivator 4(NCOA4), CDGSH iron-sulfur domain 1, heat shock protein family A (Hsp70) member 5, and acyl-CoA synthetase long chain family member 4, regulate ferroptosis in digestive system cancer [35]. The abnormal increase in ferritin is associated with other digestive system tumors, excluding biliary and esophageal tumors [35]. Therefore, understanding the relationship between ferroptosis and cancer of the digestive system may provide a better idea for treating cancer in clinics. Here, this study reviews the relationship between digestive diseases and ferroptosis from the perspectives of gastric cancer (GC), CRC, pancreatic cancer (PDAC), hepatocellular cancer (HCC), and hepatic fibrosis (Figure 1) (Table 1).

#### 2.2.1. Gastric Cancer (GC)

GC was once thought to have little to do with ferroptosis. However, the increasing evidence indicated there were relationships between gastric cancer and ferroptosis. There is a study that suggested that cisplatin and paclitaxel promote miR-522 secretion from cancer-associated fibroblasts by activating the USP7/hnRNPA1 axis, leading to arachidonate lipoxygenase 15 suppression and decreased lipid-ROS accumulation in cancer cells, and ultimately resulting in decreased chemo-sensitivity [37]. Furthermore, Hao et al. observed that cysteine dioxygenase 1 (COD1) can restore GSH activity, reverse the induction ferroptosis caused by erastin, and inhibit ROS production and LP [38]. Perilipin2 (PLIN2) is also known as an adipose differentiation-related protein (ADRP), which is involved in assisting the storage of neutral lipids within the lipid droplets [38]. Sun et al. used RNA-seq to analyze PLIN2 and differentially expressed genes modulated by PLIN2 in neoplastic tissues of both PLIN2 overexpression and knockdown groups in vivo, which indicated that overexpression and knockdown of PLIN2 augmented the proliferation and apoptosis of GC cell lines SGC7901 and MGC803, respectively, and PLIN2 was an indispensable gene and protein in the suppression of ferroptosis caused by abnormal lipomatabolism in GC [39]. Therefore, they believed that PLIN2 likely serves not only as a diagnostic biomarker but also as a new therapeutic target [39]. Gao et al. confirmed that actinidia chinensis planch could effectively prevent gastric cancer via activating apoptosis, promoting ferroptosis, and suppressing the mesenchymal phenotype [40]. By reviewing the literature, we find limited research on the link between GC and ferroptosis, and with the available information, we can foresee better treatment of GC in the future by discovering new exosomes and chemotherapeutic agents to modulate ferroptosis.

#### 2.2.2. Pancreatic Cancer (PDAC)

PDAC is a fatal malignancy of the digestive tract, and understanding the relationship between pancreatic cancer and ferroptosis maybe provide new ideas to extend the longevity of patients [35]. There are many compounds that influence the level of Fe in the cell to adjust ferroptosis. Wei et al. treated the MIN6 cell (originating from a mouse insulinoma cell line) with NaAsO2 and found that it induced ferroptosis. In this study, mitochondrial ROS-dependent autophagy, by regulating the iron homeostasis and arsenic-diminished mitochondrial membrane potential, reduced the cytochrome c level and the production of mitochondrial ROS to damage mitochondria [41]. Zhu et al. conducted a study on PDAC and reported that genetic or pharmacologic inhibition of the HSPA5-GPX4 pathway enhanced gemcitabine sensitivity by disinhibiting ferroptosis in vitro and subcutaneously and orthotopically, which hinted that HSPA5 may be a potential target against pancreatic cancer [42]. In another study, Chen et al, found that under exposure to erastin and RSL3, the inducers of ferroptosis induced ω-6 PUFA-mediated production of 4-hydroxy-2-nonenal, thus promoting ferroptosis, whereas the inhibition or silencing of arachidonate 15-lipoxygenase decreased both erastin-induced and RSL3-induced ferroptosis in pancreatic cancer [43]. Nils Eling et al. demonstrated that ART specifically induced ROS and lysosomal iron-dependent cell death in PDAC [44]. To explore the occurrence of ferroptosis in human islet transplantation, Bruni et al. treated human islets with erastin or RSL3. Analysis showed that islets were indeed susceptible to ferroptosis in vitro, and the induction of ferroptosis affects islet function, which can be recovered by administering ferroptosis inhibitors. Pancreatic cancer is notorious for its high mortality rate, and ferroptosis can be a friend to mankind if it can be activated clinically by targeting ferroptosis to inhibit cancer cell proliferation.

#### 2.2.3. Colorectal Cancer (CRC)

In the past, E2H2 and Wnt/β-catenin have been considered classical modulators of the signaling pathways in colorectal cancer [45], and recent studies have found a certain correlation between colorectal cancer and ferroptosis. Previous studies investigating the role of ferroptosis in colon cancer have primarily focused on TP53 [35]. As we all know, P53 is an evolutionarily conserved protein involved in many cellular life processes, such as cell proliferation, differentiation and death, and metabolism [46]. Jiang et al. show that P53 inhibits cystine uptake and sensitizes cells to ferroptosis, a non-apoptotic form of cell death, by repressing the expression of SLC7A11, a key component of the cystine/glutamate antiporter [47]. Therefore, TP53 is a potential target in treating different cancers by modulating ferroptosis. In another study, Shen et al. found that resibufogenin can inhibit CRC cells growth and tumorigenesis by triggering ferroptosis in a GPX4 inactivation-dependent manner [48]. Sui et al. treated HCT116 cells, LoVo cells, and HT29 CRC cells with ferroptosis inducer RSL3 for 24h and observed that ROS levels and transferrin expression were elevated in CRC cells treated with RSL3 whereas the expression of GPX4 was reduced. These findings indicate iron-dependent cell death, ferroptosis, implying that the induction of ferroptosis contributed to RSL3-induced cell death in CRC cells [49]. CRC is a highly prevalent cancer that takes a large number of lives each year, and the existing research offers the possibility of fighting CRC by activating ferroptosis.

#### 2.2.4. Hepatocellular Cancer (HCC)

Research shows two categories of genes, including ferroptosis up-regulated factors (FUF) and ferroptosis down-regulated factors (FDF), which induce and suppress ferroptosis by affecting the synthesis of GSH [50]. FUF is controlled by one transcription factor called hypermethylated in cancer 1 (HIC1), while FDF is controlled by another transcription factor named hepatocyte nuclear factor 4 alpha (HNF4A). Therefore, disrupting the balance between HIC1 and HNF4A can provide a novel approach for the treatment of HCC [50]. Sorafenib was approved by the FDA as a first-line drug for advanced HCC in 2005 [51]. One of its roles is to inhibit the initiation of translation mediated by the rapamycin kinase signaling pathway, which contributed to the inhibition of ferroptosis in HCC [35]. NRF2 is a vital regulator of the antioxidant response, whose overexpression inhibits apoptosis and contributes to chemoresistance in several cancers, and Sun et al. found that the status of NRF2 is a key factor that determines the therapeutic response to ferroptosis-targeted therapies in HCC cells, while the p62-Keap1-NRF2 pathway protects against ferroptosis in HCC cells [52,53,54]. Metallothionein-1G is a critical negative regulator of ferroptosis and has been a promising therapeutic target of sorafenib resistance in human HCC cells [55]. In other studies, low-density lipoprotein-docosahexaenoic acid promotes lipid peroxidation, GSH consumption, and reduced GPX4 activity to induce ferroptosis [35,56]. In addition, low-density lipoprotein-docosahexaenoic acid promotes lipid-peroxidation, GSH consumption, and reduces GPX4 activity, thus inducing ferroptosis.

#### 2.2.5. Hepatic Fibrosis

Liver fibrosis is a progression of chronic liver disease, which lacks effective therapies across the world [57]. Sui et al. found that heme oxygenase-1-mediated hematopoietic stem cell (HSC) ferroptosis is required in magnesium isoglycyrrhizinate to ameliorate CCl4-induced hepatic fibrosis [57]. Another study suggested that ferroptosis played an essential role in triggering inflammation in steatohepatitis [58]. Kong et al. treated HSC with artesunate, and the results revealed that artesunate remarkably promoted ferroptosis of activated HSC, which suggested ferritinophagy-mediated HSC ferroptosis was responsible for artesunate-induced anti-fibrosis efficacy, providing new clues for further pharmacological study of artesunate [59].

### 2.3. Respiratory System

The study and application of ferroptosis in the respiratory system are still in its infancy, but its double-edged role has intrigued researchers. In order to help understand whether ferroptosis is friendly to the body or not, we discuss the different effects of ferroptosis on lung diseases by introducing radiation lung injury and non-small cell lung cancer in the hope of providing help for the treatment of lung diseases.

#### 2.3.1. Radiation Lung Injury (RILI)

RILI is one of the most common complications of chest radiation therapy, associated with an increase in the risk of illness and death in patients with pneumonia [60,61]. Radiation therapy induces the production of many inflammatory cytokines and ROS that induce cellular ferroptosis [62,63]. A study by Li et al. found that the administration of ferroptosis inhibitor reduces the level of ROS in mice with acute RILI, which indicates that ROS may be the upstream factor triggering ferroptosis in acute RILI. After laser irradiation, mitochondrial atrophy was evident in the experimental group. At the same time, the downregulation of GPX4, a critical protein that regulates ferroptosis, is detected. This condition is assumed to be an important feature of ferroptosis [64,65]. Ferroptosis inhibitors can reverse this phenomenon, suggesting that ferroptosis is involved in the formation of acute RILI.

#### 2.3.2. Non-Small Cell Lung Cancer (NSCLC)

NSCLC is the most common lung cancer, accounting for about 85% of lung cancer cases. The cisplatin (CDDP) chemotherapy regimen has become the standard adjuvant therapy for patients with advanced NSCLC, but CDDP also has significant resistance [66]. Studies have shown that CDDP can lead to cell death by increasing the level of intracellular ROS, disrupting homeostasis, and triggering oxidative stress [67]. Therefore, Li et al. hypothesized that the application of ferroptosis inducers is an ideal way to solve CDDP resistance in NSCLC based on previous investigations. It was found that the activation level of System Xc- induced by CDDP was negatively correlated with the sensitivity of NSCLC cells to CDDP. When the System Xc- gene is knocked out, cell sensitivity to CDDP is significantly enhanced. After treatment of CDDP resistant cells with erastin and sorafenib, the two ferroptosis inducers induced ferroptosis in CDDP-resistant NSCLC cells by regulating System Xc-. When CDDP is combined with ergot or sorafenib, the cell survival rate is significantly reduced, and the ROS level is increased. This phenomenon can be reversed by deferoxamine (DFO) treatment, which further proves that the type of cell death is ferroptosis. In general, erastin/sorafenib induced the ferrpotosis of CDDP resistant NSCLC cells by inhibiting System Xc- [68].

### 2.4. Nervous System

As an important factor causing ferroptosis, iron plays a vital role in the metabolism of the nervous system, which efficiently induces oxidative stress in the brain and leads to ferroptosis [69]. This phenomenon has also been proven to be closely related to neurodegenerative diseases. However, not all ferroptosis in the nervous system is unfriendly, such as glioma, so we introduce glioma along with neurodegenerative diseases.

#### 2.4.1. Parkinson’s Disease (PD)

PD is the second-most common age-related degenerative disease. A lesion is located in the midbrain nucleus and substantia nigra, containing considerable amounts of iron and dopamine (DA). The presence of iron and DA may indicate that the disease is inextricably linked to ferroptosis [70]. Some studies have shown that the level of ferritin can be increased significantly after the level of neuron-like cells or neurons are treated by ferric ammonium citrate (FAC) [71]. Therefore, Zhang et al. treated dopaminergic neuroblasts with different concentrations of FAC to simulate iron overload in PD to study the mechanism of ferroptosis in PD. The results showed that the mitochondria decreased, and the density of the mitochondrial membrane increased after FAC treatment. Moreover, these changes can be reversed by the ferroptosis-specific inhibitor Fer-1. Moreover, FAC treatment resulted in decreased GPX4 levels and increased ROS in cells, both of which are associated with surface ferroptosis in PD. Since it has been reported that ferroptosis can be induced by MAPK and p53 pathways, Zhang et al. observed the phosphorylation of the MAPK pathway associated with ERK (JNK and p38) and p53 after treatment with FAC [72,73,74]. The results showed that the phosphorylation of ERK, JNK, and p38 had not changed significantly, but the phosphorylation level of p53 had increased. More evidence indicated that p53 induces cellular ferroptosis by regulating GSH metabolism and ROS levels [47]. Altogether, these results indicate that ferroptosis is involved in the production of PD via the p53 pathway.

#### 2.4.2. Alzheimer’s Disease (AD)

AD is the most common cause of dementia, accounting for about 75 percent [75]. It has previously been reported that AD amyloid deposits in senile plaques and hyperphosphorylation lead to neurofibrillary tangles [76]. At the same time, abnormal secretion of the amyloid β peptide (Aβ) in nerve cells is increased, or abnormal clearance is made [77]. These pathological changes will eventually kill neurons and even lead to death. However, lowering amyloid levels in the brain did not improve or slow AD progression, suggesting that something else was inducing AD [78]. A significant increase in iron levels in the brains of people with AD presents the possibility of ferroptosis [79]. In AD, the expression levels of Ferritin heavy chain (FTH) and Ferritin light chain (FTL) are elevated, indicating an increase in unstable iron [80]. In addition, an increase in FTL is associated with a decrease in GPX4, which means the brain has less antioxidant capacity [81]. The ferritin in AD brains can catalyze the Fenton reaction to increase brain oxidation [82]. Some studies have found that neurodegeneration was reduced in mice with degenerative diseases when given high levels of vitamin E or the ferroptosis inhibitor Liproxstatin-1 [83].

#### 2.4.3. Glioma

Glioma is the most common primary malignant intracranial tumor in adults [84]. Patients with glioblastoma have a short average life span and are highly resistant to multiple combination therapies [85]. Several studies have suggested that glioma is associated with suppressed ferroptosis [86]. Therefore, Cheng et al. further investigated the effect of ferroptosis on glioma [87]. Their study showed that the GPX4 expression level significantly increased, whereas 12-HETE and 15-HETE expression levels decreased in human glioma tissues and cells. Significantly, ferroptosis is reduced in gliomas. ACSL4, a key mediator of ferroptosis, has also shown decreased expression in gliomas [5,87]. After overexpression of ACSL4, the GPX4 expression was significantly reduced, whereas the expression of ferroptosis indicators 5-HETE, 12-HETE, and 15-HETE significantly increased, thus inhibiting the proliferation of glioma cells. Meanwhile, siRNA-mediated knockdown of ACSL4 can mediate the increase in LDH mediated by Sorafenib. More directly, the decline in cell survival induced by sorafenib could be reversed by siRNA-mediated knockdown of ACSL4. In general, the ACSL4/GPX4 pathway can inhibit the proliferation of glioma cells by activating ferroptosis, thus providing a new idea for the treatment of glioma. However, the relevant mechanisms require further study.

### 2.5. Other Systems

#### 2.5.1. Melanoma

Epidemiological statistics in the United States showed the increasing incidence of melanoma [88]. The overall five-year survival rate for melanoma has improved significantly over the past few decades, but the association between melanoma and ferroptosis is less reported. Because of this, Luo et al. conducted a series of studies that eventually identified the critical regulator of ferroptosis in melanoma cells: miRNA-137. After screening with transfected miRNA, miRNA-137 reversed erastin-and RSL3-induced ferroptosis, and cell survival was significantly improved compared with the control group. At the same time, the miRNA-137 knockdown increases erastin and RSL 3-induced cell death in melanoma cells. On this basis, the overexpression of miRNA-137 inhibits MDA levels, one of the final products of lipid peroxides induced by erastin and RSL3, and miRNA-137 can also inhibit the production of lipid ROS and Fe^2+^ [89]. Gln, a vital nitrogen source of cancer cells, has been reported to reduce oxidizable film lipids to avoid ferroptosis by inhibiting decomposition [90,91]. Luo et al. observed that Gln is necessary for miRNA-137 to inhibit ferroptosis. Furthermore, miRNA-137 inhibits Gln uptake by targeting solute-linked carrier family A1 member 5 (SLC1A5). This conclusion can be confirmed by reversing miRNA-137-mediated ferroptosis by the overexpression of SLC1A5 [91]. Therefore, it is not difficult to infer that the low expression of miRNA-137 can enhance the sensitivity of melanoma cells to erastin and provide an additional choice for the treatment of melanoma.

#### 2.5.2. Doxorubicin (DOX)-Induced Cardiomyopathy (DIC)

DIC is chronic progressive cardiomyopathy induced by DOX, a chemotherapy drug with cardiotoxicity, that has a poor prognosis and can even cause death [92]. Some studies have found that DOX-induced oxidative stress in mitochondria is one of the causes of DIC. DIC can be improved when certain antioxidant enzymes, such as manganese superoxide dismutase (Mn-SOD) and glutathione peroxidase1 (GPX1), are overexpressed [93,94]. GPX4 expression in the cytoplasm and mitochondria of cardiomyocytes significantly decreased in the DIC model of mice. After Fer-1 treatment, the level of lipopolysaccharide (LP) was significantly reduced, which improved DOX-induced cell death to a certain extent. The above phenomena suggest that ferroptosis is involved in the formation of DIC. Further research revealed that DOX functions in the mitochondria rather than outside them. DOX enters mitochondria and down-regulates the GPX4 expression level inside. After DOX enters the mitochondria, Fe^3+^ can be obtained directly by ferritin [95]. Further research revealed that DOX is at work in the mitochondria rather than outside them. DOX enters the mitochondria and down-regulates GPX4 expression (Figure 2). Fe3+ sequesters DFO whereas Dexrazoxane (DXZ) reduces iron levels in mitochondria after DOX treatment; however, it does not prevent LP accumulation and cellular ferroptosis. Surprisingly, DOX-induced LPs and cell death were significantly blocked after treatment with the Fe^2+^-specific chelating agent Mito-Ferro Green (MFG), demonstrating that mitochondrial chelation of Fe^2+^ avoided cellular ferroptosis [96]. In conclusion, DOX induces the death of myocardial iron and causes DIC by down-regulating the GPX4 level and producing the DOX-Fe^3+^ complex with Fe^3+^ after the reduction of DOX-Fe^2+^ and the production of lipid peroxides.
biomolecules-11-01790-t001_Table 1Table 1The molecules and mechanisms related to ferroptosis in different diseases.NumbersDiseasesMoleculesMechanisms1AKIHO-1HO-1 resistance to ferroptosis, and the specific cause remains to be studied [17]2PCADECR1DECR1 participates in redox homeostasis by regulating the balance between saturated and unsaturated phospholipids [30]3GCCDO1CDO1 activates GSH to inhibit the production of ROS and LP [38]4PDACcytochrome cDecreased mitochondrial membrane potential down-regulated cytochrome c and ROS levels [41]HSPA5HSPA5-GPX4 pathway [42]5CRCGPX4Inhibition of GPX4 increases ROS and transferrin [49]6HCCGSHGSH regulates ferroptosis by HIC1/HNF4A regulating FUF/FDF [50]NRF2p62-Keap1-NRF2 pathway [53]7Hepatic fibrosisHO-1Improvement of hepatic fibrosis by HO-1 induction of HSC ferroptosis [58]8RILIGPX4Down-regulation of GPX4 increases ROS production [65]9NSCLCSystem Xc-Induced the ferroptosis of CDDP-resistant NSCLC by inhibiting System Xc- [68]10PDP53P53 induces ferroptosis by regulating GSH metabolism and ROS levels [47]11ADGPX4The decrease in GPX4 content increases FHL and decreases the antioxidant capacity of brain [81]12GliomaACSL4 and GPX4Inhibit ferroptosis by inhibiting ACSL4 to increase the production of GPX4 [87]13MelanomamiRNA-137MiRNA-137 inhibits the production of MDA, ROS, and Fe^2+^ [89]14DICGPX4 and LPDOX down-regulates GPX4 and produces LP while reducing Fe^3+^ to Fe^2+^ [95]15RCCHippoHippo pathway effector TAZ regulates ferroptosis sensitivity in RCC [25]


## 3. Ferroptosis Inducers

### 3.1. Sorafenib

Sorafenib is a multi-target, multi-kinase, ATP competitive inhibitor and can also be used as a ferroptosis inducer. The specific mechanism of ferroptosis is incomplete, but studies have shown that intracellular ferroptosis is mainly regulated by System Xc- and GPX4 [97,98]. Considered a protease that inhibits LP, GPX4 degrades hydrogen peroxide and other typical small molecular peroxides and complex lipid peroxides. GSH is an indispensable cofactor in the activation process. Under normal circumstances, GPX4 prevents ferroptosis by inhibiting the accumulation of lipid peroxide in cells. Still, when GPX4 is inhibited, it can lead to the accumulation of ROS in cells, thus inducing ferroptosis [99].

Sorafenib-induced ferroptosis is independent of its poly-kinase activity but leads to the accumulation of intracellular ROS by inhibiting System Xc-, and then triggers ferroptosis. Compared with the pro-apoptosis effect dependent on its kinase inhibitory activity, sorafenib promotes ferroptosis in tumor cells [100]. Studies have shown that after exposure to sorafenib, the Rb protein function of HCC cells is lost, which promotes the ferroptosis of cancer cells [101].

The effect of sorafenib has been confirmed by clinical trials. As the first approved systemic drug for advanced liver cancer, sorafenib was approved by the FDA as the first-line drug for advanced liver cancer in 2005, offering a desirable effect [51]. In August 2009, sorafenib tablets (trade name: Nexavar) officially entered the Chinese liver cancer treatment market and have been used for the treatment of patients with advanced liver cancer who could not be operated on.

### 3.2. Erastin

In 2003, Dolma et al. used large-scale screening experiments to explore the killing effects of various compounds on cancer cells. They discovered and named the new compound eradicator of RAS and ST (erastin) [102]. Erastin as a small-molecule compound was originally discovered as an inducer of ferroptosis accompanied by the depletion of antioxidants caused by the inhibition of System Xc- [103]. Erastin reprograms cancer cell metabolism by modulating System Xc- to trigger ferroptosis [1]. Yagoda et al. discovered erastin could change the permeability of the mitochondrial outer membrane, and VDACs are the target of erastin [104]. Erastin can open up VDAC [72], which will increase the production of ROS and cause ferroptosis [105]. It can also trigger multiple molecules that differ from other ferroptosis inducers [106].

### 3.3. Sulfasalazine (SAS)

SAS also induces ferroptotic cancer cell death by the inhibition of System Xc- [107]. The ferroptotic cancer cell death induced by SAS is caused by LP and increased intracellular free iron [108]. SAS is a potential therapeutic drug for breast cancer, and Yu et al. observed that SAS can cause ferroptosis in breast cancer cells, especially in cells with a low expression of estrogen receptors [109].

KIM E H et al. reported that CISD2 inhibition can overcome the resistance of head and neck cancer (HNC) to ferroptosis induced by SAS by increasing the accumulation of mitochondrial ferrous and lipid ROS [107]. As a ferroptosis inducer, SAS has been approved for clinical use by the Food and Drug Administration (FDA) in the United States, with significant effects.

### 3.4. Ras-Selective Lethal Small Molecule 3 (RSL3)

In cancer cells, the small molecular compound RSL3 requires the accumulation of iron-dependent lipid ROS to promote ferroptosis [43,110]. RSL3 is an effective ferroptosis inducer that depends on the activity of GPX4 [49]. RSL3 triggers ferroptosis by inactivating the GPX4 [111]. As a negative target of RSL3, GPX4 mediates the inhibition of ferroptosis [49]. Ye et al. observed the combination of the ferroptosis inducer RSL3 and a low concentration of PTX can induce the death of ferroptosis cells in mtp53 hypopharyngeal squamous cell cancer cells, up-regulating the expression of the p53 protein [112]. Wang et al. noted that RSL3 inhibited the survival of glioma cells and induced glioma cell death in a dose-dependent manner [113]. Shintokur et al. reported that LOX-mediated lipid peroxide production enhances RSL3-induced ferroptosis [43]. These show that RSL3 is an effective ferroptosis inducer.

## 4. Ferroptosis Inhibitors

### 4.1. Ferrostatin-1 (Fer-1)

High-throughput screening efforts have determined that Fer-1 is an effective inhibitor of iron prolapse, and its activity lies in its capability to slow down the accumulation of lipid hydrogen peroxide [114]. Fer-1 is a radical-trapping antioxidant (RTAs) superior to vitamin E [114]. Fer-1 is more effective than phenolic antioxidants in inhibiting Ferroptosis [114]. Michael C Horwath et al. showed that the ferroptosis inhibitor Fer-1 has unexpected antifungal activity different from its anti ferroptotic activity [115]. Some studies also show that Fer-1 can block pathological cell death in the brain, kidney, and other tissues [116].

### 4.2. Baicalein

Baicalein (also known as 5,6,7-trihydroxyflavone) is a flavonoid obtained from the Scutellaria baicalensis Georgi, Lamiaceae plants originally. Qin et al. observed that baicalein can alleviate oxidative damage of neurons and inhibit ferroptosis of nerve cells by inhibiting LP [117]. Baicalin reduces the production of 4-hydroxynonenal, the mRNA expression of prostaglandin-endoperoxide synthase 2, and the expression of 12/15-LOX, which is an essential enzyme of LP, and up-regulates the expression of GPX4 in post-traumatic epilepsy (PTE) [118]. These results further prove that baicalein plays a neuroprotective role in PTE, depending on its inhibition of the ferroptosis process mediated by 12/15-LOX [117].

During the screening of the natural product library of ferroptosis inhibitors, Xie et al. observed that baicalein, acting as a natural ferroptosis inhibitor, not only inhibits ferroptosis induced by ergotamine in PDACr cells, but also has more significant anti-ferroptosis activity compared with other ferroptosis inhibitors, such as Fer-1, lipostatin-1, DFO mesylate, and β-mercaptoethanol [119]. At the biochemical level, baicalein can inhibit the production of Fe^2+^ induced by ergotamine, GSH consumption, and LP [119]. At the protein level, baicalein inhibits the degradation of GPX4 induced by erastin and protects cells from membrane LP [49].

As a flavonoid inhibitor of 12-LOX, baicalein prevents the increase in ROS by up-regulating Nrf2 and inhibiting 12-LOX, thus protecting cells and reducing the damage induced by hydrogen oxide. Baicalein has a phenolic hydroxyl group and three hydroxyl groups; it exhibits free radical scavenging activity, which selectively increases the content of H_2_O_2_ in tumor cells and causes tumor cell death. Baicalein, as a natural ferroptosis inhibitor, can not only reduce ROS and inhibit LP, but also regulate iron homeostasis, inhibit iron accumulation and complex Fe^2+^, and protect GPX4. In addition, its inhibitory effect is significantly better than that of Fer-1 and DFO mesylate [119]. Accordingly, the members of Labiatae have been widely used in clinics.

### 4.3. Others

Dixon et al. noted that ferroptosis in cells can be inhibited by iron-chelating agents, such as DFO, DFO mesylate, 2,2’-pyridine, ciclopirox olamine (CPX), and iron ions [1]. The iron-chelating agents intercept ferri ions from cytolysosome through endocytosis, thereby preventing the generation of lipid ROS [118]. Other studies have also shown thaliproxstatin-1, D-α-tocopherol, vitamin E, lysosomal activity inhibitors (lostoxin A), trolox, 2,6-ditertbutyl-4-methylphenol (BHT), pepstatin methyl ester, and ammonium chloride (NH_4_Cl) can inhibit ferroptosis by inhibiting LP [120]. Puerarin and baicalein have similarities in neuroprotective effects. The difference is that puerarin plays an important role, depending on the cytotoxicity of glutamic acid-induced Y-79 cells, in inhibiting ROS production and preventing Ca^2+^ influx. We have compiled a list of related ferroptosis inducers and inhibitors [121] (Table 2).

## 5. Conclusions

The purpose of this review is to summarize the association of ferroptosis with digestive, urinary, respiratory, neurological, and other systemic diseases, as well as to summarize some of the inducers associated with ferroptosis and the clinical progress of these inducers. Ferroptosis is an iron-dependent, novel form of cell death characterized by the loss of GPX4 activity and lipid peroxidase deposition. The concentration of intracellular iron and LP are the main biochemical characteristics of ferroptosis [132]. The pathogenesis of digestive, urinary, respiratory, neurological, and other systemic diseases is related to ferroptosis. At the same time, we have also reviewed other diseases that are not part of the four systems, but we will not go into detail here. GPX4, System Xc-, the P-53 pathway, and the ACSL4 pathway are all closely related to disease and ferroptosis. NRF2 has been tested for ferroptosis by targeting [132].

In terms of drugs related to ferroptosis, we reviewed most of the inhibitors and inducers that have been well studied in relation to ferroptosis. Erastin, sulfasalazine, RSL3, and Sorafenib are related inducers. Among them, Sorafenib, a first-line drug approved by the FDA for advanced liver cancer, has been clinically used to intervene in the treatment of ferroptosis. Fer-1, Baicalein, and iron-chelating agents are inhibitors of ferroptosis. At the end of the article, we have compiled a list of related ferroptosis inducers and inhibitors. Although, the pathogenesis of epigenetic factors controlling ferroptosis in cancer and other diseases is not fully understood, ferroptosis can still be mediated by mediating potential targets and intervening axes. Perhaps further research into ferroptosis in the future will lead to more cures for disease.

## Figures and Tables

**Figure 1 biomolecules-11-01790-f001:**
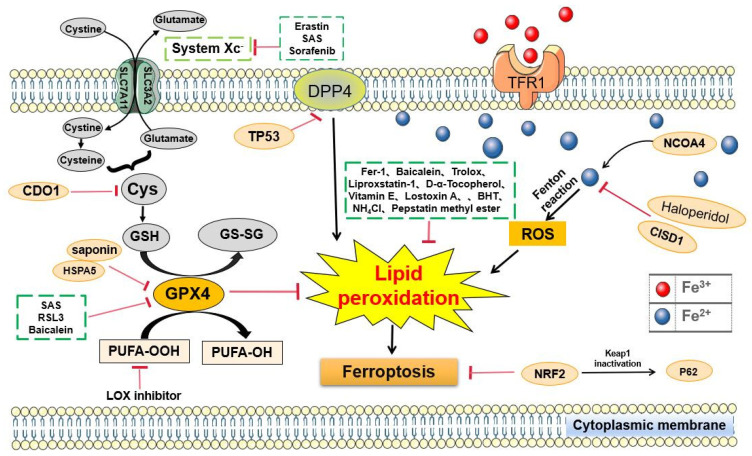
Signaling pathways regulating ferroptosis in cancer of the digestive system. An increase in human CDO1 activity is expected to decrease cellular cysteine levels, which may decrease the synthesis of GSH and promote ferroptosis. TP53 inhibits erastin-stimulated ferroptosis in a transcription-independent manner by inhibiting DPP4 in CRC. CISD1 regulates intracellular iron metabolism to prevent mitochondrial damage during ferroptosis and reduce ferroptosis and LP in liver cancer. The activation of the P62-Keap1-NRF2 pathway protects liver cancer cells from ferroptosis. LOX is used to accelerate ferroptosis.

**Figure 2 biomolecules-11-01790-f002:**
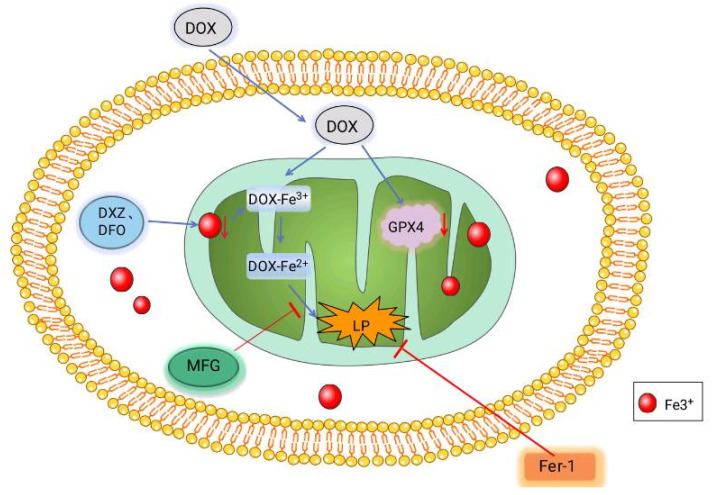
Mechanism of DOX-induced ferroptosis in cardiomyocytes. After DOX enters the mitochondria, it combines with Fe^3+^ and is reduced to DOX-Fe^2+^ to produce lipid peroxides. Meanwhile, DOX can also coordinate the level of GPX4 to promote cell ferroptosis. Fe^3+^ sequesters DXZ and DFO reduce the iron content in mitochondria. The Fe^2+^-specific chelating agent MFG blocks DOX-induced LPS and Fer-1 directly reduced LP, thereby reducing cell death and decreasing cell ferroptosis.

**Table 2 biomolecules-11-01790-t002:** Ferroptosis inducers and inhibitors.

Compounds or Drugs	Structures	Mechanisms
Erastin	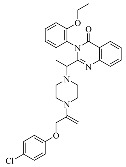	Induced ferroptosis by inhibiting System Xc- [1].
Imidazole-ketone- erastin	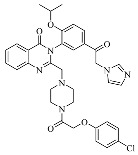	Induced ferroptosis by inhibiting System Xc- [122].
SAS	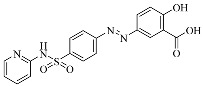	Induced ferroptosis by inhibiting GPX4 and System Xc- [123].
RSL3	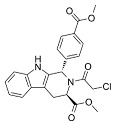	Induced ferroptosis by inhibiting GPX4 [124].
Sorafenib	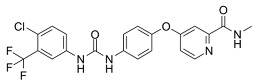	Induced ferroptosis by inhibiting System Xc- [125].
FINO2	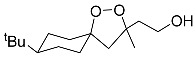	Induced ferroptosis by causing widespread lipid peroxidation including both indirect loss of GPX4 enzymatic function and directly oxidizes iron.
FIN56	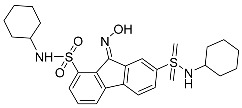	Induced ferroptosis by inducing post-translational GPX4 protein degradation [126].
Ferric ammonium citrate	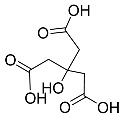	Induced ferroptosis by increasing iron abundance [2].
DPI compounds	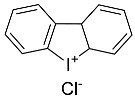	Induced ferroptosis by inhibiting System Xc- through depleting GSH [9].
Tert-butylhydroperoxide		Induced ferroptosis by inhibiting System Xc- through depleting GSH [127].
Fer-1	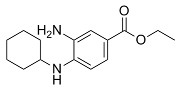	Inhibited ferroptosis by inhibiting LP [2].
Baicalein	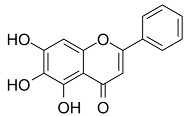	Inhibited ferroptosis by inhibiting the accumulation of iron, LP, and GPX4 degradation [119,128].
DFO mesylate	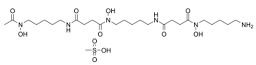	Inhibited ferroptosis by inhibiting the accumulation of iron.
2, 2’-pyridine	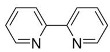	Inhibited ferroptosis by inhibiting the accumulation of iron [1].
CPX	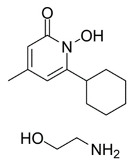	Inhibited ferroptosis by inhibiting the accumulation of iron [5].
Liproxstatin-1	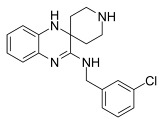	Inhibited ferroptosis by inhibiting LP [129].
D-α-Tocopherol	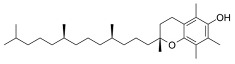	Inhibited ferroptosis by inhibiting LP.
Vitamin E	C_29_H_50_O_2_	Inhibited ferroptosis by inhibiting LP [130].
Lostoxin A		Inhibited ferroptosis by inhibiting LP.
Trolox	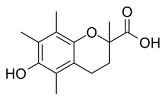	Inhibited ferroptosis by inhibiting LP.
BHT	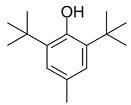	Inhibited ferroptosis by inhibiting LP.
Pepstatin methyl ester		Inhibited ferroptosis by inhibiting LP.
NH4Cl		Inhibited ferroptosis by inhibiting LP [131].
Puerarin	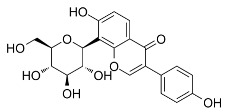	Inhibited ferroptosis by inhibiting ROS production and Ca^2+^ influx [1].

## Data Availability

Not applicable.

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
