# Peer review of "Insight into the Double-Edged Role of Ferroptosis in Disease"

_biomolecules, 2021, doi:10.3390/biom11121790_

Round 1
Reviewer 1 Report
It is an interesting review which intends to show the role of ferroptosis in several diseases. However, the main problem of this review, in my opinion, is the small number of references on each subject directly involved with ferroptosis. For example, 5 references for acute kidney injury, 5 for chronic kidney disease and renal tumors, 2 for prostate cancer, 4 for gastric carcinoma, 2 for hepatocellular carcinoma, 3 for hepatic fibrosis, and so on. This greatly diminishes the usefulness of this article as an adequate review of the subject.
However, I believe that, if presented in another way, it could be interesting. For example, instead of covering all diseases, the article could focus on cancer, a subject on which most references are provided.
The part related to drugs for promoting and inhibiting ferroptosis is good. I believe a text restricted to cancer and drugs (in this section some examples related to other diseases could be provided) would be much more convincing.
Reviewer 2 Report
The manuscript by Zhang et al. extensively summarized the literature about ferroptosis-related diseases. In the first part, the authors reviewed the involvement of ferroptosis-related molecules in different types of disease. Then they introduced the ferroptosis inducers and inhibitors, as well, the action mechanisms in disease.
1. Many ferroptosis regulators have been linked to the progression of the disease which is driven by unknown or complex mechanisms. The authors are suggested to make a simple table to summarize the associated ferroptosis molecules in different diseases.
2. Please add the references of identified mechanism exerting by ferroptosis-regulating agents in Table 1.
3. Please label the direction of the arrow in Figure 1 system Xc- to show the cystine in and glutamine out. “Figure 1” label is missing in context.
4. Please give the full name of molecule abbreviations such as CRK (in section 2.1.2.), ATTs (in section 2.1.3.), ART (in section 2.2.2.), FHL (FHL, section 2.4.2.)
5. In the conclusion section, what is NRE2? Or is NRF2?
6. There are many spelling errors in the manuscript. Please carefully check and correct them.
Round 2
Reviewer 1 Report
The authors have substantially improved the text.
Author Response
Thank you very much for participating in the peer review process of this manuscript.